# Validation of the Barthel Index as a Predictor of In-Hospital Mortality among COVID-19 Patients

**DOI:** 10.3390/healthcare11091338

**Published:** 2023-05-06

**Authors:** Julio Alberto Mateos-Arroyo, Ignacio Zaragoza-García, Rubén Sánchez-Gómez, Paloma Posada-Moreno, Ismael Ortuño-Soriano

**Affiliations:** 1Department of Pneumology, Hospital General Nuestra Señora del Prado, 45600 Talavera de la Reina, Spain; 2Department of Nursing, Faculty of Nursing, Physiotherapy and Podology, University Complutense of Madrid, 28040 Madrid, Spain; 3Instituto de Investigación Sanitaria Hospital 12 de Octubre (imas12), 28041 Madrid, Spain; 4FIBHCSC, Instituto de Investigación Sanitaria Hospital Clínico San Carlos (IdISSC), 28040 Madrid, Spain

**Keywords:** nursing assessment, COVID-19, mortality, activities of daily living, validation study

## Abstract

In order to predict the high mortality due to COVID-19, simple, useful and remote instruments are required. To assess the validity of the baseline Barthel Index score as a predictor of in-hospital mortality among COVID-19 patients, a validation study of a clinical prediction tool in a cohort of patients with COVID-19 was conducted. The primary variable was mortality and the Barthel Index was the main explanatory variable. Demographic, clinical and laboratory variables were collected. Other mortality predictor scores were also assessed: Pneumonia Severity Index, CURB-65 and A-DROP. The Receiver Operating Characteristic Area under the Curve (ROC AUC), sensitivity and specificity were calculated for both the Barthel Index and the other predictor scores. An analysis of the association between the main variables was conducted, adjusting by means of three multivariate models. Three hundred and twelve patients were studied. Mortality was 16.4%. A mortality Odds Ratio (OR) of 5.95 was associated with patients with a Barthel Index ≤ 90. The model number 3 was developed to predict in-hospital mortality before COVID-19 infection occurs. It exhibits an OR of 3.44, a ROC AUC of 0.792, a sensitivity of 74.5% and a specificity of 73.9%. The Baseline Barthel Index proved useful in our population as a predictor of in-hospital mortality due to COVID-19.

## 1. Introduction

Between January and December 2019, the World Health Organisation (WHO) confirmed 41 cases of an infection by a new coronavirus, called SARS-CoV-2, in the city of Wuhan, China [1,2]. On 11 March 2020, the WHO reported 118,000 cases (4291 deaths) in 114 countries and characterised the situation as a pandemic [3]. The spread of the virus continued and one year later, in March 2021, the number of cases had risen to 116,736,437 and deaths to 2,593,285 worldwide [4].

The disease, referred to as COVID-19, presents with a fever, sometimes with dyspnoea and bilateral pulmonary infiltrates [5]. Since the beginning of the pandemic, several studies sought to establish the typical profile of patients affected by COVID-19 [6], as well as clinical and analytical indicators that could be predictors of mortality due to this disease [7]. One of the first findings was the association between advanced age and mortality due to COVID-19. The meta-analysis by Zheng et al. [7] found a six-fold increased risk in patients over 65 years of age.

At present, there are validated tools for estimating mortality in different types of pneumonia, including SARS-CoV-2-related pneumonia [8], such as the Pneumonia Severity Index (PSI) [9], the CURB-65 [10] score and the A-DROP score [11]. The disadvantage of these tools is that one must wait for the individual to be admitted to hospital for different invasive diagnostic tests to be carried out. The six-minute walk test has recently been validated as a predictor of COVID-19 mortality [12], but in a situation of social distancing, it would be ideal to have a simple tool that could be obtained remotely [13].

On the other hand, different authors state that frailty generally has a greater effect than age and comorbidities on COVID-19-related mortality [14], which suggests that impaired physical functioning and the performance of basic and routine activities prior to a SARS-CoV-2 infection could be predictive of the risk of mortality even before an individual acquires the disease.

The Barthel Index (BI), or Maryland Disability Index, determines patients’ independence in their basic activities of daily living (BADL) [15]. It is a widely used tool due to its reliability and validity [16]. In addition, the BI has been used for purposes other than those originally established. It has been used as a predictor of mortality in patients admitted to geriatric units [17] and to predict mortality from community-acquired pneumonia (CAP) in the general population [18], independently or in combination with the PSI [19]. The aim of our study was to analyse whether a baseline Barthel Index (pre-infection) could predict in-hospital mortality in patients with COVID-19 in adult academic hospitals and to compare it with other currently validated scores.

## 2. Materials and Methods

### 2.1. Study Design

A prospective observational validation study in a cohort of adult patients admitted with COVID-19 to a public third-level hospital.

### 2.2. Sample/Participants

Patients ≥ 18 years old diagnosed with COVID-19 pneumonia who were admitted to hospital wards were included. Patients were excluded if they had a positive PCR (Polymerase Chain Reaction) test or were positive for antigens, but did not have a radiological diagnosis of pneumonia.

Based on the incidence of in-hospital mortality in the SEMI-COVID-19 registry (17.5% in a sample of 15,111 patients) [20], a confidence level of 95%, an accuracy of 4.50 units and an expected loss of 0.5%, the minimum sample size was estimated to be 276 patients. The replacement rate proved to be higher in the end, so the recruitment time was prolonged so that the statistical power would not be decreased.

### 2.3. Data Collection

The data were collected by nurses from hospital wards service. To avoid systematic errors in the data collection and the application of the scores, the healthcare professionals who collaborated on the data collection were previously instructed and trained. Recruitment was conducted between November 2020 and February 2021.

#### 2.3.1. Primary Outcome

The primary outcome was to identify the incidence of the in-hospital mortality among adult COVID-19 patients and assess their prognosis on the basis of the basal BI.

The variable in-hospital mortality due to COVID-19 was obtained by following up each patient during their entire hospital stay; its outcome was dichotomised (yes/no).

The BI is obtained by asking 10 questions, the answers to which lead to a score between 0 and 100 points. The higher the score, the greater the independence in the performance of BADL [15]. In addition, different degrees of dependency can be established: total dependence (0–20 points), severe dependence (21–60 points), moderate dependence (61–90 points), slight dependence (91–99 points), and independent (100 points) [21].

The BI was applied as soon as possible after the COVID-19 diagnostic with the patients being oriented to answer based on their experiences before the diagnostic. If the patient had cognitive impairment or problems with effective communication, the BI was assessed through an interview of a close relative or primary caregiver [22].

#### 2.3.2. Secondary Outcome

The secondary outcome was to compare the prediction of COVID-19 mortality by the pre-infection BI to other validated tests that require invasive testing (PSI, CRUB-65 and A-DROP).

The PSI is composed of 19 items of a different nature: sociodemographic parameters, comorbidities and analytical parameters [9]. Higher scores indicate a higher probability of death.

CURB-65 is an acronym for the parameters that comprise the instrument: Confusion, Urea, Respiratory rate, Blood pressure and age > 65. Each of these parameters is assessed dichotomously. Each question scores 1 point, so the result ranges from 0 to 5 points. Increasing scores indicate an increased risk of mortality [10].

The A-DROP tool is a modification of CURB-65 and is very similar to its parent scale. It is also evaluated based on a score ranging from 0 to 5 points [11].

#### 2.3.3. Variables

The following variables were analysed:

(a) Patient-related variables: sociodemographic variables (age, gender and institutionalised), comorbidities (arterial hypertension, cardiovascular disease, diabetes, dyslipidaemia, COPD, asthma, active neoplasia, liver disease, chronic kidney disease, cerebrovascular disease and Charlson Index), physical examination data (temperature, blood pressure, heart rate, oxygen saturation and confusional state), laboratory data (lymphocytes, haematocrit, D-dimer, glucose, urea, sodium, ferritin, C-Reactive Protein, arterial pH and arterial pO2) and days of hospital stay.

The Charlson comorbidity index is a marker for chronic disease burden that helps predict the ten-year mortality for a patient who may have a range of comorbid conditions. The scores range from 0 to 33, with higher scores indicating a greater burden of chronic illness.

The confusional state was estimated by qualitative perception by healthcare workers. Three questions were asked: Do you know what day it is today? Do you know where you are? Do you know what happened to you? If the patient fails in all questions, he/she is considered confused.

All these variables were collected at the time of patient admission except for this last one.

(b) Index and scores: during the hospital stay, the following scores for estimating COVID-19 mortality were collected: PSI, CURB-65 and A-DROP. In addition, the baseline BI was obtained prior to hospital admission.

### 2.4. Ethical Considerations

Approval from the ethical and clinical research committee of the centre was obtained (Code 30/2020).

### 2.5. Data Analysis

Categorical variables were expressed as a frequency and percentage. The chi-square (χ^2^) or Fisher’s test was used for between-group comparisons. Normality was tested using the Kolmogorov–Smirnov test. Quantitative variables were expressed as the mean and standard deviation for normal distributions and median and interquartile range for non-normal distributions, and groups were compared using the Student’s *t*-test or Mann–Whitney test, as appropriate. Odds Ratios (ORs) with their respective confidence intervals were also obtained.

The sensitivity, specificity, positive and negative likelihood ratios (LR+ and LR−) and accuracy were calculated for the best cut-off points of the BI, PSI, CURB-65 and A-DROP. Their respective Receiver Operating Characteristic (ROC) curves were plotted and the different Areas Under the Curve (AUCs) with their confidence intervals were compared.

For the multivariate logistic regression analysis, all patient-related variables described above were used except for days of hospital stay. The variables were entered separately together with the BI and the variation of the model was observed. For the design of the possible models, the following criteria were followed: accuracy (OR for the BI with the narrowest 95% confidence intervals); no requirement for invasive techniques to obtain the result of the variable; and the consideration of socio-demographic factors, including at least age and gender. The final validation of the models was carried out using the Hosmer–Lemeshow goodness-of-fit test, and the predicted probabilities were represented by the ROC curves and their AUCs.

Some of the continuous variables were dichotomized. Their cut-off point was set using the Youden index [23] for the maximization of sensitivity and specificity in the classification of the primary outcome.

Individuals with missing data in their variables that prevented the calculation of all predictor scores (BI, PSI, CURB-65 or A-DROP) were excluded from the analysis.

For all comparisons, a statistical significance level of *p* < 0.050 was established. The data were analysed using the IBM SPSS Statistics software Version 24.0.

### 2.6. Validity and Reliability/Rigour

All values and behaviours were measured by scoring items from the International Council of Nurses Code of Ethics for Nurses, 2021 revised English version.

## 3. Results

The data of 312 patients were analysed. A flow diagram with the patients’ progression through the study is provided in the Figure 1.

### 3.1. Patient Characteristics

The mean age of the patients was <65 years, the majority of them (54.2%) were men. Thirty-six patients (11.5%) were institutionalised prior to hospital admission and the median Charlson comorbidity index score was 1 [0–2]. The most prevalent chronic disease was hypertension with 53.8% (*n* = 168), followed by dyslipidaemia, cardiovascular disease and diabetes with 39.4% (*n* = 123), 27.9% (*n* = 87) and 23.4% (*n* = 73), respectively. Regarding clinical findings, values were within the normal range, except for a reduction in mean oxygen saturation of 91.26 ± 5.93 percent and a partial oxygen pressure of 68.68 ± 22.77 mm Hg. In addition, there was an increase in D-dimer with 2892.04 ± 7559.34 ng/mL and C-reactive protein levels of 115.44 ± 165.38 mg/L.

### 3.2. Primary Outcome

In-hospital mortality was 16.4% (51/312) [95% CI: 12.6%–20.9%]. The deceased group had a significantly higher age, Charlson Index and urea levels than the non-deceased group (79.40 ± 11.60 years vs. 66.60 ± 15.80 years; 1 [1–2] vs. 0 [0–1]; 81.00 ± 49.18 mg/dL vs. 48.07 ± 33.93 mg/dL, respectively). In addition, they showed a significantly higher percentage of institutionalised and confused patients (31.4% vs. 7.7% and 21.6% vs. 8.8%, respectively). On the other hand, the deceased group presented significantly lower values of oxygen saturation (88.4% vs. 91.8%; *p* < 0.001) and arterial pO2 (61.17 ± 17.25 mmHg vs. 70.18 ± 23.46 mmHg; *p* = 0.010) on admission (Table 1). The Barthel score was significantly lower in the deceased group (75 [30–100] vs. 100 [90–100]; *p* < 0.001), as is the case with the rest of the recorded indexes (Table 1).

The BI has an area under the ROC curve of 0.736. The optimal cut-off point of the BI was 90 points, with a sensitivity and specificity of 70.6% and 71.3%, respectively (Table 2 and Figure 2). Individuals with a BI ≤ 90 points show a 5.95 times higher risk of death than those without baseline functional impairment (Table 2).

To analyse the effect of other variables on the prediction of BI, three multivariate models were generated by a logistic regression; in all of them, the BI was adjusted for other regression covariates. In all models, the confidence intervals for the OR of the BI are narrower than the unadjusted estimate [95% CI: 3.08–11.51] and therefore improve the accuracy of this indicator.

Model 1, consisting of BI, PSI and SatO2, had the highest AUC-ROC (0.804), followed by model 3 (0.792), which consists of BI, age, gender and Charlson Index (Table 3 and Table 4). Furthermore, model 1 had the highest sensitivity with 78.4% and model 3 had the highest specificity in the analysis (73.9%). A correct classification of the BI and all the models was observed in terms of the adequate categorisation into the two possible states, surviving or deceased, with values ranging from 83.0 to 84.0% (Table 4).

### 3.3. Secondary Outcome

The predictive ability of the BI is similar to that of the other indexes, and even higher than that of CURB-65 (Table 2). PSI had the best discriminative ability, although A-DROP and BI follow with curves that overlap at some points (Figure 2). With the exception of CURB-65, all instruments have an AUC-ROC above 0.700. The tool with the highest sensitivity was CURB-65 (88.2%), showing a specificity of 31.0% (Table 2). The validity of the PSI for its best cut-off point was 72.5% and 76.2% for sensitivity and specificity, respectively. Except for the CURB-65 score, all instruments exceeded the value of 2 for LR+. As for the LR- for the best cut-off points, all of them were below 0.400, which corresponds to the BI (Table 2).

## 4. Discussion

All analyses performed in this study indicate that the BI is an adequate predictor of in-hospital mortality due to COVID-19. A BI of 90 was established as the cut-off point for predicting the likelihood of in-hospital mortality.

### 4.1. Sample Characteristics

The sample studied showed characteristics similar to other multicentre studies with large samples. Laosa et al. [24] describe similar data with a median age of 66.06 ± 15.33 years and 55.2% men. Casas-Rojo et al. [20] reported slightly higher data for the median age (69.40 [56.40–79.90] years) and men (57.2%). They also describe similar data for the prevalence profile of chronic diseases: 50.9% were hypertensive, 39.7% had dyslipidaemia and 19.4% had diabetes mellitus. Therefore, the sample collected in this study may be representative of the profile of patients admitted to Spanish hospitals.

Mortality in our study was 16.4% [95% CI: 12.6–20.9%]. This is lower than that reported in other multicentre studies. Docherty et al. [25] describe a mortality of 26.0% in the UK, Richardson et al. [26] find 21.0% in the New York City Area and Laosa et al. [24] and Casas-Rojo et al. [20] report 19.8% and 21.0%, respectively, in Spain. On the other hand, an Italian study described a mortality of 7.2% [27], which is lower than ours.

In relation to the above, it should be added that some authors have found that being male carries an increased risk of mortality from COVID-19 infection [28,29]. The studies carried out in New York and the UK include more men than in our study (60.3% and 59.9% respectively), and both include patients admitted directly to the ICU, who may have a higher mortality rate due to greater severity. On the other hand, all the studies discussed above were carried out in the first stage of the pandemic and therefore prior to ours, with the clinical management still unknown. On the other hand, the Italian study, although carried out at the beginning of the pandemic and with a higher percentage of men (70%), shows a lower mortality than all of them. When reviewing their results, they stratify mortality by age, with 20.2% in those over 80 years of age. Some authors claim that increasing age increases in-hospital mortality due to COVID-19 [30,31]. On the other hand, a meta-analysis published by Wang et al. [32] describes that the number of deaths attributed to COVID-19 has been disparate in different countries.

### 4.2. The BI as a Predictor of In-Hospital Mortality among COVID-19 Patients

The mean BI in the deceased indicates severe dependence, whereas the mean BI in survivors indicates moderate dependence (*p* < 0.001). Moreover, the same is true when the BI is analysed in a dichotomised manner using 90 as the cut-off point (≤90 vs. >90). Although the BI was not designed with this objective in mind, it yielded, without adjustment for other covariates, an AUC-ROC of more than 0.700, as well as a very acceptable sensitivity and specificity. We consider these results suitable for two reasons: (a) as stated by Swets [33], a ROC AUC between 0.700 and 0.900 is useful for some specific purposes, and tests with values > 0.900 are considered to have high accuracy; and (b) the BI obtained a similar AUC as the reference scores PSI, CURB-65 and A-DROP. Separately, none of the scores analysed for predicting in-hospital mortality due to COVID-19 achieved an AUC of more than 0.900, and none were clearly better than any of the other ones. The likelihood ratios and accuracy of the BI showed an expected behaviour, with low scores in the deceased and high scores in the survivors. The results confirm that the BI is an adequate predictor of in-hospital mortality. The interest of this result is related to the fact that the BI is simple to calculate, does not require invasive measures and can be completed remotely, e.g., by telephone [34,35]. This makes it possible to screen patients without the need for them to visit a health centre. On the other hand, it is important to note that it is a measure that is routinely obtained in many health centres to monitor patients, mainly for disabilities in the elderly [36,37]. This suggests that healthcare staff are more familiar with obtaining this index than the PSI, CURB-65 and A-DROP.

### 4.3. Adjusting the BI for Prediction of In-Hospital Mortality Due to COVID-19

In our analysis, age was shown to be particularly important. At the end of the follow-up, the mean age of the deceased was close to 80 years. In contrast, our results did not show statistically significant differences in gender, although the mortality was higher in the men’s group (62.7% vs. 37.3%; *p* = 0.170). In spite of this, we considered it important to adjust the BI for gender, although in most models, significance was lost and the fit did not improve.

Despite the low comorbidity indicated in both groups by the Charlson Index, it is significantly higher in the deceased group. Therefore, adjusting the model for this variable may have helped to dilute the confounding effect of patients with high comorbidities.

On the other hand, although patient institutionalisation showed significance in the bivariate analysis, it led to a confounding behaviour in relation to age, as more than 90% of them were more than 75 years old.

We were unable to find a model that combined all three criteria defined in the methodology: maximum accuracy, avoidance of invasive techniques and inclusion of the variables gender and age together, and therefore three different models were generated. The model that showed the highest accuracy was the one that fitted BI with the PSI scale and SatO2 (model 1), with an OR of 2.65 [95% CI: 1.24–5.66]. The combination of BI and PSI has been used elsewhere to estimate the mortality risk in CAP [19], with an improved accuracy compared to using each separately. In the present study, it also led to an improvement in the adjustment of the prediction of COVID-19 pneumonia mortality. Model 2 was able to adjust the BI with covariates that did not require invasive techniques: age and SatO2. Finally, although model 3 was the least accurate model, it allowed in-hospital mortality to be established exclusively with baseline variables prior to a COVID-19 infection. The great advantage of the latter model is that it allows establishing a patient’s risk before a hypothetical hospital admission. This may be important for the adoption of different strategies, such as prioritisation in vaccination programmes or a selection of individuals for booster doses of the vaccine against COVID-19.

### 4.4. AUC, Sensitivity and Specificity of Models

All models increased the AUC, sensitivity and specificity values compared to the BI alone. Model 1 was the best model: its AUC exceeded 0.8 and it had 78.4% sensitivity and 72.4% specificity. Model 2 had the highest percentage of correctly classified individuals, with an accuracy of 84.0%. Model 3 showed the highest specificity of the three models (73.9%) for correctly classifying survivors. This is important, as it is the only model that can be applied with pre-infection data. Another noteworthy aspect of model 3 is that the probability of mortality in men shows an OR = 2.01, while keeping the rest of the covariates, including the BI, fixed. This is consistent with other studies where mortality in men was found to be higher than in women [7], something we were unable to confirm in the unadjusted analysis.

In the practical application of the different models, it should be noted that: Model 1, while providing the best results, is the most complex to obtain. It requires the use of bloody methods. It also requires the calculation of the PSI. Model 2 is an improvement over model 1, PSI, A-DROP and CURB-65, in that it is not necessary to carry out any bloody method. The only thing that is necessary is that the patient goes to a health centre to obtain the SatO2 or has access to a pulse oximeter at home to measure peripheral saturation. Model 3 is perhaps the one with the greatest advantages over the other models and scales (PSI, A-DROP and CURB-65). This model can be calculated using pre-infection data. In addition, it is not necessary to use bloody methods with the patient and all values can be obtained remotely (e.g., BI via telephone) [34,35].

### 4.5. The BI as a Screening Tool and Its Relationship with Other Prediction Models

The BI has been used as a screening tool for community-acquired respiratory diseases before the COVID-19 pandemic [18,19]. Recently, researchers found an association between BI and COVID-19 mortality. They described that a five-point decrease in the BI increases mortality by 10.0–15.0% [24]. However, they did not specify at what point the BI was obtained—at hospital admission or prior to infection, as was the case in our study. On the other hand, in order to obtain the risk of a patient, one must wait until the patient is infected and hospitalised, and it is not possible to predict mortality before infection, something that our model does allow. Similarly, Wang et al. [38] and da Costa et al. [39] carried out a validation study of the Barthel Index as a predictor of mortality, but in both studies they collected the value of the BI at admission.

Fumagalli et al. [40] developed a prediction model in which they included previous frailty and the BI 2 weeks before infection in institutionalised elderly people ≥ 75 years of age. Although the results they present are acceptable (AUC = 0.870), they use the COVID-19 Mortality Risk Score, a series of variables defined by the authors that are subsequent to the patient’s infection and that must be collected on hospital admission.

The combination of risk estimation and an analysis of diagnostic/predictive validity is one of the strengths of this study. We did not find other studies comparing validity (AUC, sensitivity, etc.) specifically for the BI as a tool for predicting COVID-19 mortality.

With respect to the data obtained for the reference indices for mortality prediction (PSI, CURB-65 and A-DROP), our results are in agreement with other studies [8,41,42,43,44]. The high discriminative power of the PSI may be due to the fact that age and gender are included in the index itself, with a downward correction for younger ages and women. Therefore, models using the BI + PSI should not be further adjusted for these factors.

## 5. Limitations

The observational nature of this study is limited to obtaining associations and it cannot establish causality between functional status and mortality. In addition, this is a single-centre study and therefore there may be bias due to population characteristics; although, as noted, the study population is similar to other cohorts used in multicentre studies. We did not analyse the impact of body weight or other variables like the BMI due to the impossibility of obtaining reliable measures of them. Neither was the variable “admission to the ICU” or those patients who were admitted to the ICU from the beginning. At the time of data collection for this study, only patients requiring invasive mechanical ventilation were admitted to the ICU. However, these patients were mainly younger and had a good baseline functional situation. Even though it is highly probable that admission to the ICU could have a bias or impact on mortality, our main result was adjusted by age in our multivariate models 2 and 3. Finally, it should be noted that the use of dichotomous variables simplified the analysis to improve understanding, at the cost of a possible loss of information.

## 6. Conclusions

We found a strong association between baseline patient functional status and in-hospital mortality due to COVID-19. The BI can be considered a valid score to determine this relationship, showing that only with this instrument and other non-invasive variables prior to admission (age, gender and the Charlson Index), the risk of death in the event of a COVID-19 infection requiring hospital admission can be estimated. These results allow us to discriminate the population that is most likely to die in hospital from COVID-19 before becoming infected, which could help us to make individualised prevention decisions about the general population.

## Figures and Tables

**Figure 1 healthcare-11-01338-f001:**
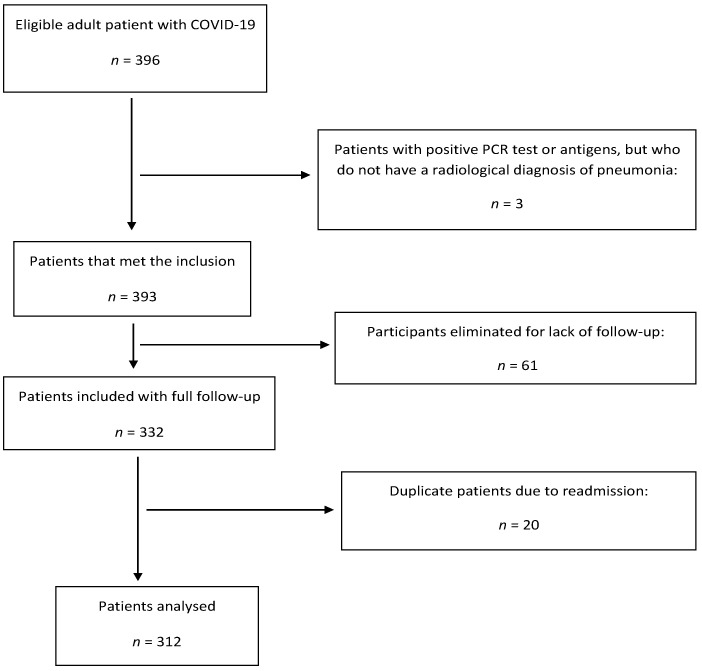
Flow Diagram Showing Patients’ Movement through the Study.

**Figure 2 healthcare-11-01338-f002:**
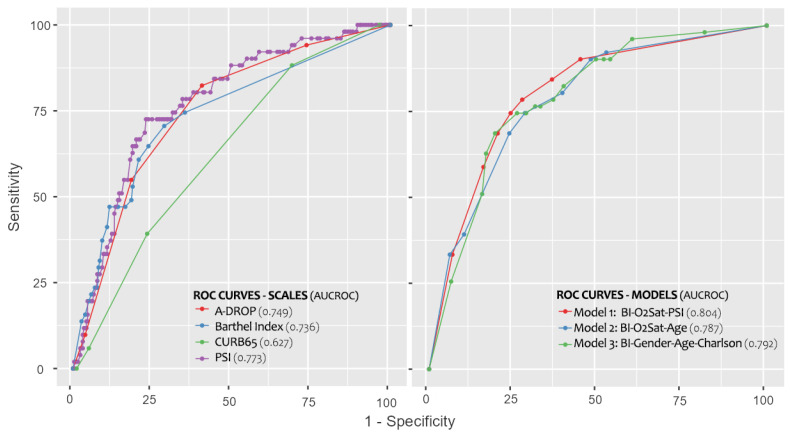
ROC Curves for the Predictor Scores and the Multivariate Models. (PSI: Pneumonia Severity Index; BI: Barthel Index; O2Sat: oxygen saturation).

**Table 1 healthcare-11-01338-t001:** Characteristics of the Patients Studied and Comparison of Sociodemographic, Clinical and Laboratory Findings Between Survivors and Deceased.

Variables	Total (*n* = 312)	Survivors (*n* = 261)	Deaths (*n* = 51)	*p*
**General**				
Age in years, mean ± SD	68.72 ± 15.87	66.63 ± 15.77	79.41 ± 11.57	<0.001 ^f^
Men, n (%)	169 (54.2)	137 (52.5)	32 (62.7)	0.170 ^g^
**Clinical history**				
Charlson Index ^a^, median [IR]	1 [0–2]	0 [0–1]	1 [1–2]	<0.001 ^h^
Nursing home resident, n (%)	36 (11.5)	20 (7.7)	16 (31.4)	<0.001 ^g^
Chronic diseases, n (%) ^b^				
Hypertension	168 (53.8)	131 (50.2)	37 (72.5)	0.003 ^g^
Cardiovascular disease	87 (27.9)	64 (24.5)	23 (45.1)	0.003 ^g^
Diabetes	73 (23.4)	60 (23.0)	13 (25.5)	0.700 ^g^
Dislipidaemia	123 (39.4)	98 (37.5)	25 (49.0)	0.120 ^g^
Chronic obstructive pulmonary disease	22 (7.1)	21 (8.0)	1 (2.0)	0.145 ^g^
Asthma	39 (12.5)	33 (12.6)	6 (11.8)	0.860 ^g^
Active neoplasia	14 (4.5)	9 (3.4)	5 (9.8)	0.060 ^g^
Liver disease	9 (2.9)	8 (3.1)	1 (2.0)	0.990 ^g^
Nephropathy	18 (5.8)	12 (4.6)	6 (11.8)	0.910 ^g^
Cerebrovascular disease	29 (9.3)	19 (7.3)	10 (19.6)	0.014 ^g^
**Clinical Variables on admission**				
Temperature (°C), mean ± SD	36.92 ± 0.96	36.91 ± 0.94	36.99 ± 1.10	0.600 ^f^
Blood pressure (mmHg)				
Systolic, mean ± SD	139.26 ± 27.32	138.33 ± 27.54	143.98 ± 25.92	0.180 ^f^
Diastolic, mean ± SD	77.36 ± 16.12	77.60 ± 16.80	76.00 ± 12.11	0.510 ^f^
Heart rate (beats per minute), median [IR]	92 [82–104]	94 [82–104]	87 [81–102]	0.147 ^h^
% Oxygen saturation, mean ± SD	91.26 (5.93)	91.82 (5.70)	88.41 (6.26)	<0.001 ^g^
Tachypnoea–RR ≥ 30, n (%) ^b^	71 (22.8)	60 (23.0)	11 (21.6)	0.049 ^g^
Pleural effusion, n (%) ^b^	17 (5.4)	14 (5.4)	3 (5.9)	0.747 ^g^
Patients with Confusion, n (%) ^c^	34 (10.9)	23 (8.8)	11 (21.6)	0.007 ^g^
**Analytical values**				
Lymphocytes (×10,000/µL), mean ± SD	1.06 ± 0.88	1.09 ± 0.93	0.91 ± 0.59	0.180 ^f^
Haematocrit %, mean ± SD	39.80 ± 5.85	40.13 ± 5.70	38.13 ± 6.43	0.020 ^f^
D-Dimer (ng/mL), mean ± SD ^d^	2892.04 ± 7559.34	2237.41 ± 4945.23	6537.18 ± 15,097.28	0.068 ^f^
Glucose (mg/dL), mean ± SD	154.15 ± 77.80	149.36 ± 64.92	178.67 ± 122.46	0.100 ^f^
Urea (mg/dL), mean ± SD	53.46 ± 38.73	48.07 ± 33.93	81.00 ± 49.18	<0.001 ^f^
Sodium (mmol/L), mean ± SD	138.45 ± 4.90	138.37 ± 4.80	138.86 ± 5.40	0.510 ^f^
Ferritin (ng/mL), mean ± SD ^e^	603.64 ± 558.20	584.98 ± 546.55	709.20 ± 617.92	0.230 ^f^
C-Reactive Protein (mg/L), mean ± SD	115.44 ± 165.38	112.29 ± 176.30	131.58 ± 90.37	0.450 ^f^
pH (Arterial), mean ± SD	7.46 ± 0.05	7.46 ± 0.48	7.46 ± 0.62	0.790 ^f^
Arterial partial oxygen pressure pO2 (mmHg), mean ± SD	68.68 ± 22.77	70.18 ± 23.46	61.17 ± 17.25	0.010 ^f^
**COVID-19 mortality risk indices**				
Barthel index ^b^, median [IR]	100 [80–10]	100 [90–100]	75 [30–100]	<0.001 ^h^
Independent (BI = 100)	182 (58.3%)	169 (92.9%)	13 (7.1%)	<0.001 ^g^
Slight dependence (BI = 91–99)	19 (6.1%)	17 (89.5%)	2 (10.5%)	0.479 ^g^
Moderate dependence (BI = 61–90)	44 (14.1%)	32 (72.7%)	12 (27.3%)	0.035 ^g^
Severe dependence (BI = 21–60)	37 (11.9%)	25 (67.6%)	12 (32.4%)	0.005 ^g^
Total dependence (BI = 0–20)	30 (9.6%)	18 (60.0%)	12 (40.0%)	<0.001 ^g^
Pneumonia Severity Index (PSI), median [IR]	84 [63–116]	77 [61–105]	122 [95–144]	<0.001 ^h^
CURB-65, median [IR]	2 [1–3]	2 [1–2]	2 [2–3]	0.002 ^h^
A-DROP, median [IR]	1 [1–2]	1 [0–2]	2 [2–3]	<0.001 ^h^

Note. Statistically significant values in bold (*p* < 0.050); ^f^ Student’s *t*-test; ^g^ Chi-Squared test or Fisher’s exact test; ^h^ Mann–Whitney test; SD: Standard Deviation; IR: Interquartile Range; BI: Barthel Index; RR: Respiratory Rate. ^a^ The Charlson comorbidity index is a marker for chronic disease burden that helps predict the ten-year mortality for a patient who may have a range of comorbid conditions. The scores range from 0 to 33, with higher scores indicating a greater burden of chronic illness. ^b^ Dichotomous variables (yes/no), expressing the percentage of its occurrence. ^c^ Qualitative perception by healthcare workers. Three questions were asked: Do you know what day it is today? Do you know where you are? Do you know what happened to you? If the patient fails in all questions, he/she is considered confused. ^d^
*n* = 289. ^e^
*n* = 233.

**Table 2 healthcare-11-01338-t002:** Validity of the Discriminative Ability for COVID-19 Mortality of the Different Predictor Scores.

Score	Cut-Off Point	AUC ^a^ (95% CI)	*p*	Sens ^b^% (95% CI)	Spec ^c^% (95% CI)	Correct Classification ^d^% (95% CI)	LR+ ^e^ (95% CI)	LR− ^f^ (95% CI)	OR ^g^ (95% CI)
Barthel	90	0.736 (0.657–0.816)	<0.001	70.6 (57–81)	71.3 (65–76)	71.2 (66–76)	2.46 (1.89–3.19)	0.41 (0.27–0.64)	5.95 (3.08–11.51)
PSI ^h^	107	0.773 (0.705–0.840)	<0.001	72.5 (59–82)	76.2 (71–81)	75.6 (71–80)	3.05 (2.32–4.02)	0.36 (0.23–0.57)	8.48 (4.31–16.71)
A-DROP	2	0.749 (0.678–0.819)	<0.001	82.4 (70–90)	59.4 (53–65)	63.1 (58–68)	2.03 (1.67–2.46)	0.30 (0.16–0.55)	6.82 (3.19–14.61)
CURB-65	2	0.627 (0.549–0.705)	0.004	88.2 (77–94)	31.0 (26–37)	40.4 (36–46)	1.28 (1.12–1.46)	0.38 (0.17–0.82)	3.38 (1.38–8.23)

^a^ AUC: Area under the curve. ^b^ Sens: Sensitivity. ^c^ Spec: Specificity. ^d^ Correct classification: Accuracy or efficiency of the test. ^e^ LR+: Positive likelihood ratio. ^f^ LR−: Negative likelihood ratio. ^g^ OR: Odds Ratio. 95% CI: 95% Confidence Interval. ^h^ PSI: Pneumonia Severity Index.

**Table 3 healthcare-11-01338-t003:** Barthel Index Adjusted in the Multivariate Models.

	Variable	Categories	Reference	*p*	OR ^a^ Exp (β)	95% CI ^b^ OR Exp(β)	
						Lower	Upper
**Model 1**	Barthel Index	>90	Ref				
		≤90		0.012	2.65	1.24	5.66
	PSI ^c^	<107	Ref				
		≥107		<0.001	4.20	1.93	9.15
	SatO2	>90%	Ref				
		≤90%		0.022	2.23	1.12	4.44
**Model 2**	Barthel Index	>90	Ref				
		≤90		0.004	2.97	1.42	6.25
	Age	<75 years	Ref				
		≥75 years		0.004	3.12	1.45	6.71
	SatO2	>90%	Ref				
		≤90%		0.001	3.06	1.57	5.95
**Model 3**	Barthel Index	>90	Ref				
		≤90		0.001	3.44	1.62	7.32
	Gender	Women	Ref				
		Men		0.046	2.01	1.01	3.97
	Age	<75 years	Ref				
		≥75 years		0.021	2.49	1.14	5.43
	Charlson Index	0	Ref				
		≥1		0.035	2.37	1.06	5.27

^a^ OR: Odds Ratio. Exp(β): Exponential of beta; ^b^ 95% CI: 95% Confidence Interval; ^c^ PSI: Pneumonia Severity Index.

**Table 4 healthcare-11-01338-t004:** Areas Under the Curve with their Respective Confidence Intervals (95% CI), Sensitivity, Specificity, Correct Classification and Goodness-of-fit Test for the Models and the Barthel Index.

Model	AUC ^a^ (95% CI)	*p*	Sens ^b^%	Spec ^c^%	Correct Classification ^d^%	Hosmer Lesmeshow (*p*)
1	0.804 (0.740–0.869)	<0.001	78.4%	72.4%	83.3%	0.912
2	0.787 (0.721–0.853)	<0.001	74.5%	71.6%	84.0%	0.724
3	0.792 (0.730–0.855)	<0.001	74.5%	73.9%	83.7%	0.462
Barthel	0.736 (0.657–0.816)	<0.001	70.6%	71.3%	83.0%	-

^a^ AUC: Area Under the Curve; ^b^ Sens: Sensitivity; ^c^ Spec: Specificity. ^d^ Correct classification: Accuracy or efficiency of the test. 95% CI: 95% Confidence Interval. Cut-off value 0.500.

## Data Availability

The data presented in this study are available on request from the corresponding author.

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
