# Peer review of "Validation of the Barthel Index as a Predictor of In-Hospital Mortality among COVID-19 Patients"

_healthcare, 2023, doi:10.3390/healthcare11091338_

Round 1

Reviewer 1 Report

Dear authors, congratulations on the excellent manuscript. The information provided is beneficial for clinical practice. Notwithstanding, some points could be improved to make the manuscript clear for the readers. Below, some of these aspects are pointed out.

Main points:

-       I suggest the analysis of subgroups based on gender differences to see if the results keep similar. I also recommend that the characterization table be presented with the information for men and women separately;

-       At the beginning of the discussion, the present mortality results were compared to other studies. What can explain the difference between this manuscript and others? This explanation should be inserted in the discussion.

-       Yet, in the discussion, it is essential to discuss the possible reasons that make the Barthel index as good as other indicators used to predict mortality among COVID-19 patients. Also, it is worthwhile to show the applicability of the Barthel Index in this context.

- Other articles show the impact of body weight and body composition on COVID-19 severity, so why did you opt not to insert this variable in your analysis? If this information is available, insert this into the analysis. Otherwise, this should be addressed in the limitation section.

-       In the AUC section of the discussion, it is necessary to provide some direction to the readers about which model is better to use in clinical practice since the three models fit well. Also, it is necessary to discuss why to choose one or another. I noticed this before the limitations. However, I recommend approaching this in further detail in the AUC session in the discussion.

-       Why have the continuous variables, such as age and oxygen saturation, been approached as categorical variables? This approach must be explained in the method and discussed in the discussion session to help the readers understand the methodology and the possible limitations of the categorical approach.

-       The same is applied to the Barthel Index that was investigated based on the cutoff point, so I suggest inserting supplementary material to show the analysis with the continuous variables, specifically, the Barthel Index, Age, and Oxygen Saturation. This aspect is already shown in the limitation section, but it will be helpful to show this information as supplementary material.

Minor corrections:

-       The information about the Charlson index put in table 1 note should be inserted in the methods when the Charlson index is shown for the first time. The same should be applied to the confusing concept;

-       It is stated in the objective that Barthel Index was applied pre-infection. However, the inclusion criteria were to be diagnosticated with COVID-19. So, I think this part should be explained in the introduction in the following way “Barthel Index was applied as soon as possible after the COVID-19 diagnostic with the patients has been oriented to answer based on their experiences before the diagnostic”. The document should clarify this aspect since all the measurements were made after the patient’s admission.

-       In the abstract, it was shown that “a model is developed…”. Which model are you referring to? Is the model using just the Barthel Index as a predictor? The model needs to be approached in the abstract. Also, I think “was” is the most suitable term for this sentence instead of “is” since it is the past tense.

-       I suggest the remotion of the data related to age at the beginning of the results section since the same data is shown in Table 1 and other variables such as the Charlson index, urea, and others. This pattern happened with the Hosmer and Lemeshow goodness-of-fit test;

-       The reference list must be standardized. There are some discrepancies in the uppercases in the article’s titles.

Based on these comments, I hope you can improve the quality of the manuscript.

Author Response

Reviewer 1

Dear authors, congratulations on the excellent manuscript. The information provided is beneficial for clinical practice. Notwithstanding, some points could be improved to make the manuscript clear for the readers. Below, some of these aspects are pointed out.

 Main points:

-       I suggest the analysis of subgroups based on gender differences to see if the results keep similar. I also recommend that the characterization table be presented with the information for men and women separately;

Author: We considered the sex variable in our first analysis. However, we did not find any relevant association between sex and the primary outcome or the main scores studied. This is why we decided not to include it in the table. The only difference found was between sex and PSI. We presented in the original manuscript a possible explanation on lines 380-3825.

According to your recommendation we have made a table in which the variables are differentiated by gender. In any case, if it is considered that these data could be relevant (general analysis by sex and subgroups), they could be attached in the supplementary material.

The following table shows the analysis of the variables by sex:

Total

Deaths

Survivors

Men

Women

p-value

Men

Women

p-value

Men

Women

p-value

General

Age in years, mean±SDa

70.7±13.9

69.6±17.8

.585

78.5±10.7

81.0±13.0

.456

68.4±13.6

67.8±17.8

.782

Clinical history

Charlson Index, median [IR]b

1 [0-2]

1 [0-1]

.577

1 [0-2]

2 [1-2]

.403

0 [0-1]

0 [0-1]

.488

Nursing home, n (%)c

21 (12.4)

15 (10.5)

.722

10 (31.3)

6 (31.6)

.980

11 (8.0)

9 (7.3)

.815

Chronic diseases

Hypertension, n (%)c

98 (58.0)

70 (49.0)

.138

23 (71.9)

14 (73.7)

.889

75 (54.7)

56 (45.2)

.122

Cardiovascular disease, n (%)c

53 (31.4)

34 (23.8)

.173

13 (40.6)

10 (52.6)

.405

40 (29.2)

24 (19.4)

.065

Diabetes, n (%)c

40 (23.7)

33 (23.1)

.999

5 (15.6)

8 (42.1)

.050

35 (25.5)

25 (20.2)

.302

Dislipidaemia, n (%)c

72 (42.6)

51 (35.7)

.257

18 (56.3)

7 (36.8)

.180

54 (39.4)

44 (35.5)

.512

Chronic obstructive pulmonary disease, n (%)c

19 (11.2)

3 (2.1)

.003

1 (3.1)

0 (0)

.999

18 (13.1)

3 (2.4)

.001

Asthma, n (%)c

21 (12.4)

18 (12.6)

.999

5 (15.6)

1 (5.3)

.392

16 (11.7)

17 (13.7)

.622

Active neoplasia, n (%)c

7 (4.1)

7 (4.9)

.964

3 (9.4)

2 (10.5)

.999

4 (2.9)

5 (4.0)

.740

Liver disease, n (%)c

15 (8.9)

14 (9.8)

.935

0 (0)

1 (5.3)

.373

6 (4.4)

2 (1.6)

.286

Nephropathy, n (%)c

11 (6.5)

7 (4.9)

.542

3 (9.4)

3 (15.8)

.659

8 (5.8)

4 (3.2)

.314

Cerebrovascular disease, n (%)c

15 (8.9)

14 (9.8)

.782

8 (25.0)

2 (10.5)

.287

7 (5.1)

12 (9.7)

.156

Clinical Variables on admission

Temperature (ºC), mean±SDa

36.8 ±1.0

36.8 ±0.8

.960

36.9±0.9

36.4±0.7

.140

36.7±0.9

36.8±0.8

.477

Blood pressure (mmHg)

Systolic, mean±SDa

142.3 ±29.6

139.7 ±26.0

.476

141.0±24.6

150.9±28.5

.586

142.4±30.7

138.0±25.5

.297

Diastolic, mean±SDa

80.7 ±20.6

76.3 ±12.2

.055

73.4±11.5

78.9±12.8

.706

82.0±21.8

75.9±12.2

.031

Heart rate (beats per minute), median [IR]b

91 [81-104]

95 [82-104]

.389

88.5

[79-106.2]

84 [81-100]

.441

92

[81-104]

95

[84.2-104]

.244

% Oxygen saturation, mean±SDa

91.0 ±5.6

91.8 ±5.4

.264

88.6±5.1

88.5±6.4

.517

91.5±5.5

92.3±5.1

.326

Tachypnoea - RR≥ 30, n (%)c

44 (26.0)

27 (18.9)

.172

8 (25.0)

3 (15.8)

.505

36 (26.3)

24 (19.4)

.184

Pleural effusion, n (%)c

11 (6.5)

6 (4.2)

.518

2 (6.3)

1 (5.3)

.999

9 (6.6)

5 (4.0)

.364

Patients with Confusion, n(%)c

13 (7.7)

21 (14.7)

.048

5 (15.6)

6 (31.6)

.291

8 (5.8)

15 (12.1)

.075

Analytical values

Lymphocytes (x10,000/µL), mean±SDa

1.0 ±0.4

1.2 ±1.4

.037

0.7±0.3

1.2±0.8

.074

1.0± 0.4

1.2± 1.2

.088

Haematocrit %, mean±SDa

41.2 ±6.0

38.1 ±5.1

<.001

38.2±6.6

36.3±6.2

.019

41.5± 6.2

38.6± 4.6

<.001

D-Dimer (ng/ml), mean±SDa

3049.8 ±9116.6

2159.4 ±3866.6

.186

9072.1 ±19527.7

3646.9 ±8340.9

.247

1795.1± 3888.2

1937.86± 2663.3

.724

Glucose (mg/dl), mean±SDa

169.1 ±97.1

152.6 ±68.3

.143

216.0±161.4

173.5±106.1

.521

160.0±75.6

150.0±61.2

.205

Urea (mg/dl), mean±SDa

60.3 ±34.0

51.3 ±43.5

.082

94.1±49.7

83.3±62.8

.958

53.6± 24.9

46.8±38.3

.007

Sodium (mmol/l), mean±SDa

138.4 ±4.6

139.6 ±5.9

.078

138.9±7.2

140.3±3.9

.298

138.3±3.8

139.5± 6.2

.125

Ferritin (ng/ml), mean±SDa

761.7 ±571.9

442.3 ±501.3

<.001

857.4±432.6

541.3±795.0

.102

733.4± 593.7

427.0± 449.2

<.001

C-Reactive Protein (mg/l), mean±SDa

118.1 ±83.1

121.2 ±254.2

.900

153.6±94.5

102.1±81.7

.107

110.7± 79.6

121.5± 271.0

.880

pH (Arterial), mean±SD

7.5 ±0.1

7.5 ±0.1

.926

7.5±0.1

7.5±0.0

.976

7.5±0.0

7.5±0.0

.880

Arterial Partial Oxygen Pressure pO2 (mmHg), mean±SDa

67.3 ±19.4

72.5 ±26.7

.099

61.4±20.4

62.5±14.4

.383

68.7± 19.1

73.9± 27.9

.007

COVID-19 mortality risk indices

Barthel index, median [IR]b

100 [85-100]

100 [60-100]

.052

80 [12.5-98.7]

45 [30-100]

.716

100

[95-100]

100

[76.2-100]

.859

Independent (BI=100)c

107 (63.3)

75 (52.4)

.052

8 (25.0)

5 (26.3)

.999

99 (72.3)

70 (56.5)

.008

Slight dependence (BI=91-99)c

9 (5.3)

10 (7.0)

.539

2 (6.3)

0 (0)

.266

7 (5.1)

10 (8.1)

.334

Moderate dependence (BI=61-90)c

25 (14.8)

19 (13.3)

.703

9 (28.1)

3 (15.8)

.497

16 (11.7)

16 (12.9)

.763

Severe dependence (BI=21-60)c

10 (5.9)

27 (18.9)

<.001

4 (12.5)

8 (42.1)

.037

6 (4.4)

19 (15.3)

.003

Total dependence (BI= 0-20)c

18 (10.7)

12 (8.4)

.500

9 (28.1)

3 (15.8)

.497

9 (6.6)

9 (7.3)

.826

Pneumonia Severity Index (PSI), median [IR]b

91

[73-121]

71

[57.5-107]

<.001

129 [108.2-145.7]

117

[78-144]

.306

86[70.5-109.5]

68

[57-100.7]

<.001

CURB-65, median [IR]b

2

[1-3]

2

[1-2]

.464

2 [2-3]

2

[2-3]

.491

2

[1-2.5]

2

[1-2]

.785

A-DROP, median [IR]b

2

[1-3]

1

[1-3]

.140

3 [2-3]

2

[1-3]

.583

1

[1-2]

1

[0-2]

.305

Note. Statistically significant values in bold (p<.050); aStudent’s t-test; bMann-Whitney test; cChi-Squared test or Fisher’s exact test.

-       At the beginning of the discussion, the present mortality results were compared to other studies. What can explain the difference between this manuscript and others? This explanation should be inserted in the discussion.

Author: We agree with your suggestion. There are interesting differences between the different studies that could modify the mortality outcome.

We have added the following:

“In relation to the above, it should be added that some authors have found that being male carries an increased risk of mortality from COVID19 infection (Her AY, 2022; Fernández-Martínez NF, 2021).The studies carried out in New York and the UK include more men than in our study (60.3% and 59.9% respectively), and both include patients admitted directly to the ICU, who may have a higher mortality rate due to greater severity. On the other hand, all these studies discussed above were carried out in the first stage of the pandemic and therefore prior to ours, with clinical management still unknown. On the other hand, the Italian study, although carried out at the beginning of the pandemic and with a higher percentage of men (70%), shows a lower mortality than all of them. When reviewing their results, they stratify mortality by age, with 20.2% in those over 80 years of age. Some authors claim that increasing age increases in-hospital mortality due to COVID-19 (Maximiano Sousa F, 2021; Henkens MTHM, 2022). On the other hand, a meta-analysis published by Wang et al. (Wang H, 2020) describes that the number of deaths attributed to COVID-19 has been disparate in different countries.”

References:

Her AY, Bhak Y, Jun EJ, Yuan SL, Garg S, Lee S, Bhak J, Shin ES. Sex-specific difference of in-hospital mortality from COVID-19 in South Korea. PLoS One. 2022 Jan 24;17(1):e0262861. doi: 10.1371/journal.pone.0262861. PMID: 35073365

Fernández-Martínez NF, Ortiz-González-Serna R, Serrano-Ortiz Á, Rivera-Izquierdo M, Ruiz-Montero R, Pérez-Contreras M, Guerrero-Fernández de Alba I, Romero-Duarte Á, Salcedo-Leal I. Sex Differences and Predictors of In-Hospital Mortality among Patients with COVID-19: Results from the ANCOHVID Multicentre Study. Int J Environ Res Public Health. 2021 Aug 26;18(17):9018. doi: 10.3390/ijerph18179018.

Maximiano Sousa F, Roelens M, Fricker B, Thiabaud A, Iten A, Cusini A, Flury D, Buettcher M, Zukol F, Balmelli C, Zimmermann P, Troillet N, Vuichard-Gysin D, Schreiber PW, Bernhard-Stirnemann S, Tschudin-Sutter S, Nussbaumer-Ochsner Y, Sommerstein R, Gaudenz R, Marschall J, Senn L, Gardiol C, Keiser O, Schüpbach G, Wymann M, Vidondo B, Ch-Sur Study Group. Risk factors for severe outcomes for COVID-19 patients hospitalised in Switzerland during the first pandemic wave, February to August 2020: prospective observational cohort study. Swiss Med Wkly. 2021 Jul 28;151:w20547. doi: 10.4414/smw.2021.20547. PMID: 34324698.

Henkens MTHM, Raafs AG, Verdonschot JAJ, Linschoten M, van Smeden M, Wang P, van der Hooft BHM, Tieleman R, Janssen MLF, Ter Bekke RMA, Hazebroek MR, van der Horst ICC, Asselbergs FW, Magdelijns FJH, Heymans SRB; CAPACITY-COVID collaborative consortium. Age is the main determinant of COVID-19 related in-hospital mortality with minimal impact of pre-existing comorbidities, a retrospective cohort study. BMC Geriatr. 2022 Mar 5;22(1):184. doi: 10.1186/s12877-021-02673-1.

Wang, H.; Paulson, K.R.; Pease, S.A.; Watson, S.; Comfort, H.; Zheng, P.; Aravkin, A.Y.; Bisignano, C.; Barber, R.M.; Alam, T.; et al. Estimating Excess Mortality Due to the COVID-19 Pandemic: A Systematic Analysis of COVID-19-Related Mortality, 2020–21. The Lancet 2022, 399, 1513–1536, doi:10.1016/S0140-6736(21)02796-3.

-       Yet, in the discussion, it is essential to discuss the possible reasons that make the Barthel index as good as other indicators used to predict mortality among COVID-19 patients. Also, it is worthwhile to show the applicability of the Barthel Index in this context.

Author: Thank you for your suggestion. The following text is added to point 4.2:

“The results confirm that the BI is an adequate predictor of in-hospital mortality. The interest of this result is related to the fact that the BI is simple to calculate, does not require invasive measures and can be completed remotely, e.g. by telephone (Asuzu D, 2015; Musa KI, 2018). This makes it possible to screen patients without the need for them to visit a health centre. On the other hand, it is important to note that it is a measure that is routinely obtained in many health centres to monitor patients, mainly for disability in the elderly (Cabañero-Martínez MJ, 2009; Yang M, 2014). This suggests that healthcare staff are more familiar with obtaining this index than the PSI, CURB-65 and A-DROP.”

References:

Asuzu D, Nyström K, Schindler J, Wira C, Greer D, Halliday J, Sheth KN. TURN Score Predicts 90-day Outcome in Acute Ischemic Stroke Patients After IV Thrombolysis. Neurocrit Care. 2015 Oct;23(2):172-8. doi: 10.1007/s12028-015-0154-5.

Musa KI, Keegan TJ. The change of Barthel Index scores from the time of discharge until 3-month post-discharge among acute stroke patients in Malaysia: A random intercept model. PLoS One. 2018;13(12):e0208594. Published 2018 Dec 20. doi:10.1371/journal.pone.0208594

Cabañero-Martínez MJ, Cabrero-García J, Richart-Martínez M, Muñoz-Mendoza CL. The Spanish versions of the Barthel index (BI) and the Katz index (KI) of activities of daily living (ADL): a structured review. Arch Gerontol Geriatr. 2009;49(1):e77-e84. doi:10.1016/j.archger.2008.09.006

Yang M, Ding X, Dong B. The measurement of disability in the elderly: a systematic review of self-reported questionnaires. J Am Med Dir Assoc. 2014;15(2):150.e1-150.e1509. doi:10.1016/j.jamda.2013.10.004

-  Other articles show the impact of body weight and body composition on COVID-19 severity, so why did you opt not to insert this variable in your analysis? If this information is available, insert this into the analysis. Otherwise, this should be addressed in the limitation section.

Author: We agree with your observation. It is a limitation of the study. We add the following sentence in the section Limitations:

(lines 388-389): “We didn’t analyze the impact of body weight or other variables like the BMI due to the impossibility of obtaining reliable measures of them”.

On the other hand, Hendren et al. (Hendren NS, 2021) conducted a multicentre study in 88 hospitals in the USA, including 7606 patients. In their conclusions, they found that weight can be a risk factor for in-hospital mortality or mechanical ventilation, but mainly in young individuals (age ≤ 50 years) and with class 3 obesity (≥40 kg/m2).  Aghili et al. (Aghili SMM, 2021), carried out a meta-analysis in which they also found a relationship between obesity and COVID19 mortality, but like Hendren et al. (Hendren NS, 2021), they found that the effect increases when there is obesity >35 kg/m2 and mainly in young people. They report that obesity in elderly individuals may be accompanied by more comorbidities, for example, obesity is associated with type-2 diabetes mellitus (T2DM) that is seen frequently in those aged 65 years, an aspect that is considered in our study. We believe that, although this is a limitation of our study, it could be mitigated by the fact that the mean age of our population is 68.72±15.87 years, with a 25th percentile of 58.25 years of age.

References:

Hendren NS, de Lemos JA, Ayers C, Das SR, Rao A, Carter S, Rosenblatt A, Walchok J, Omar W, Khera R, Hegde AA, Drazner MH, Neeland IJ, Grodin JL. Association of Body Mass Index and Age With Morbidity and Mortality in Patients Hospitalized With COVID-19: Results From the American Heart Association COVID-19 Cardiovascular Disease Registry. Circulation. 2021 Jan 12;143(2):135-144. doi: 10.1161/CIRCULATIONAHA.120.051936. Epub 2020 Nov 17. PMID: 33200947.

Aghili SMM, Ebrahimpur M, Arjmand B, et al. Obesity in COVID-19 era, implications for mechanisms, comorbidities, and prognosis: a review and meta-analysis. Int J Obes (Lond). 2021;45(5):998-1016. doi:10.1038/s41366-021-00776-8

-       In the AUC section of the discussion, it is necessary to provide some direction to the readers about which model is better to use in clinical practice since the three models fit well. Also, it is necessary to discuss why to choose one or another. I noticed this before the limitations. However, I recommend approaching this in further detail in the AUC session in the discussion.

Author: Thank you very much for your comment. We agree with what you suggest. We have added to the text a description of the aspects to be taken into account in each model.

We have added in point 4.4 of the discussion the following:

“In the practical application of the different models, it should be noted that:

Model 1, while providing the best results, is the most complex to obtain. It requires the use of bloody methods. It also requires the calculation of the PSI.

Model 2 is an improvement over model 1, PSI, A-DROP, CURB-65, in that it is not necessary to carry out any bloody method. The only thing that is necessary is that the patient goes to a health centre to obtain the SatO2 or has access to a pulse oximeter at home to measure peripheral saturation.

Model 3 is per-haps the one with the greatest advantages over the other models and scales (PSI, A-DROP, CURB-65). This model can be calculated using pre-infection data. In addition, it is not necessary to use bloody methods with the patient and all values can be obtained remotely (e.g. BI via telephone)” (Asuzu D, 2015; Musa KI, 2018).

References:

Asuzu D, Nyström K, Schindler J, Wira C, Greer D, Halliday J, Sheth KN. TURN Score Predicts 90-day Outcome in Acute Ischemic Stroke Patients After IV Thrombolysis. Neurocrit Care. 2015 Oct;23(2):172-8. doi: 10.1007/s12028-015-0154-5.

Musa KI, Keegan TJ. The change of Barthel Index scores from the time of discharge until 3-month post-discharge among acute stroke patients in Malaysia: A random intercept model. PLoS One. 2018;13(12):e0208594. Published 2018 Dec 20. doi:10.1371/journal.pone.0208594

-       Why have the continuous variables, such as age and oxygen saturation, been approached as categorical variables? This approach must be explained in the method and discussed in the discussion session to help the readers understand the methodology and the possible limitations of the categorical approach.

Author: The categorization of continuous variables is common in some investigations where multivariate logistic regression models are used (Sanz F, 2018). The main reasons for the transformation are simplicity in the interpretation of the results and the establishment of "high" and "low" risks. This categorization can lead to a loss of information (Baneshi, et al.). This is why we have included this aspect in the limitations section. In any case, the strategy to minimize this loss or the introduction of some type of bias was to consider the variables individually as predictors of mortality, establishing the best cut-off points according to the Youden index (Youden WJ, 1950) that maximizes sensitivity and specificity. This means that the proportion of false positives and false negatives is minimized. Therefore, we believe that the choice of cut-off points based on this particular methodology and not on arbitrary points may have minimized a possible bias effect. We have added this aspect in the Materials and methods section. In addition, as we have reflected in the answer to your next question, there are also clinical reasons for the choice of these cut-off points.

We have added the following sentence to the methodology:

Some of the continuous variables were dichotomized. The cutoff point for them was set using the youden index for maximization of sensitivity and specificity in the classification of the primary outcome”. (Youden WJ, 1950)

References:

Sanz F,  Morales-Suárez-Varela M, Fernández E, Force L, Pérez-Lozano MJ, Martín V, Egurrola M, Castilla J, Astray, J, Toledo D, et al. A Composite of Functional Status and Pneumonia Severity Index Improves the Prediction of Pneumonia Mortality in Older Patients. Journal of General Internal Medicine 2018, 33, 437–444, doi:10.1007/s11606-017-4267-8.

Baneshi M, Talei, A. Dichotomisation of Continuous Data: Review of Methods, Advantages, and Disadvantages. Iranian Journal of Cancer Prevention 2011, 4.

Youden WJ. Index for Rating Diagnostic Tests. Cancer 1950, 3, 32–35, doi:10.1002/1097-0142(1950)3:1<32::aid-cncr2820030106>3.0.co;2-3.

-       The same is applied to the Barthel Index that was investigated based on the cutoff point, so I suggest inserting supplementary material to show the analysis with the continuous variables, specifically, the Barthel Index, Age, and Oxygen Saturation. This aspect is already shown in the limitation section, but it will be helpful to show this information as supplementary material.

Author: In your previous suggestion, we have commented on the methodology followed to establish the cut-off points for continuous variables. We would like to point out that there are some categorization issues of an exclusively clinical nature that we would like to point out:

  1. In clinical practice, Respiratory failure is defined when at rest, awake and breathing room air, arterial O2 pressure (PaO2) is less than 60 mm Hg and this value is correlated, by the haemoglobin dissociation curve, an oxygen saturation of 90% (Shebl E, 2022; Jubran A, 1990; Choong SK, 2022; González-Pozo, G, 2018).
  2. Regarding the Pneumonia Severity Score (PSI), we would like to point out that the authors who designed and validated this prognostic instrument indicated the convenience of establishing cut-off points for the establishment of 4 different mortality risk classes. (Fine MJ, 1997). These authors, as we do in our manuscript, point out the limitation of using dichotomous variables in their score. Ten years after its creation and validation, it was affirmed that because of its methodological rigor, superior prognostic accuracy, and proven effectiveness for clinical decision PSI could be consider as a gold standard for severity adjustment and risk stratification of patients with pneumonia. (Aujesky D, 2008).

Finally, in the case of age, we agree that it can be a limitation. Although it has been mentioned in the limitations section, we add to the supplementary material the calculation of the different models with age as a continuous variable.

Model 1 has no variation as age is not included.

Model 2 increases the AUC and sensitivity, decreasing the specificity.

Model

AUC (95% CI)

Sens %

Spec %

2 with age as a continuous variable

.792 (.728-.857)

86.3%

65.1%

2 with age as a dichotomous variable

.787 (.721-.853)

74.5%

71.6%

Model 3 slightly increases AUC and specificity. Sensitivity remains the same.

Model

AUC (95% CI)

Sens %

Spec %

3 with age as a continuous variable

.793 (.729-.857)

74,5%

74,3%

3 with age as a dichotomous variable

.792 (.730-.855)

74.5%

73.9%

This leads us to think that by using the variable age as a continuous variable, the result of the models does not vary excessively. We add these analyses to the supplementary material.

References:

Shebl E,  Mirabile V.S, Sankari A; Burns B. (2023, February 15). Respiratory Failure. In StatPearls. StatPearls Publishing. Retrieved April 4, 2022 from https://www.ncbi.nlm.nih.gov/books/NBK526127/

Jubran A, Tobin MJ. Reliability of pulse oximetry in titrating supplemental oxygen therapy in ventilator‑dependent patients. Chest 1990;97:1420‑5.

Choong SK. Approach to acute respiratory failure for frontline clinicians. Singapore Medical Journal. 2022; 63(12):740-745. Doi: 10.4103/singaporemedj.SMJ-2022-002

González-Pozo, G.; Santiago, A.; Lerín, M.; Iglesias, A. Acute respiratory failure. Medicine - Programa de Formación Médica Continuada Acreditado 2018, 12, 3862–3869, doi:10.1016/j.med.2018.10.020

Fine, M.J.; Weissfeld, L.A.; Kapoor, W.N. A Prediction Rule to Identify Low-Risk Patients with Community-Acquired Pneumonia. The New England Journal of Medicine 1997, 8.

Aujesky, D.; Fine, M.J. The Pneumonia Severity Index: A Decade after the Initial Derivation and Validation. Clin Infect Dis 2008, 47 Suppl 3, S133-139, doi:10.1086/591394.)

Minor corrections:

 -       The information about the Charlson index put in table 1 note should be inserted in the methods when the Charlson index is shown for the first time. The same should be applied to the confusing concept;

Author: We agree with this suggestion. We have added the description of the Charlson index and the confusional state to the methodology.

-       It is stated in the objective that Barthel Index was applied pre-infection. However, the inclusion criteria were to be diagnosticated with COVID-19. So, I think this part should be explained in the introduction in the following way “Barthel Index was applied as soon as possible after the COVID-19 diagnostic with the patients has been oriented to answer based on their experiences before the diagnostic”. The document should clarify this aspect since all the measurements were made after the patient’s admission.

Author: Thank you very much for the suggestion. We have incorporated it into the methodology to make it more understandable for the reader.

-       In the abstract, it was shown that “a model is developed…”. Which model are you referring to? Is the model using just the Barthel Index as a predictor? The model needs to be approached in the abstract. Also, I think “was” is the most suitable term for this sentence instead of “is” since it is the past tense.

Author: We apologise for the error. In the summary we refer to model 3. As discussed above, the positive aspect of this model is that it can be obtained remotely, without the need to go to a health service or to have measuring devices at home. We have clarified this in the summary. In addition, we have changed "is" to "was" to make the text more coherent.

-       I suggest the remotion of the data related to age at the beginning of the results section since the same data is shown in Table 1 and other variables such as the Charlson index, urea, and others. This pattern happened with the Hosmer and Lemeshow goodness-of-fit test;

Author: We agree that data from tables should not be repeated in the text. It is also true that age is an interesting factor in relation to COVID 19 and mortality. Taking into account your suggestion, we have reformulated the paragraph to avoid repeating data, but highlighting the issue of age in the results.

Regarding the paragraph describing the Hosmer and lesmeshow p-values, we fully agree. We have withdrawn it.

-       The reference list must be standardized. There are some discrepancies in the uppercases in the article’s titles.

Author: Thank you very much for your recommendation. We have proceeded with the revision of all references and have finished standardising them.

Based on these comments, I hope you can improve the quality of the manuscript.

Author: Thank you very much for your effort in reviewing the article. We believe that with your comments, the article has improved considerably.

Reviewer 2 Report

Please find attached the document.

Author Response

Reviewer 2

The aim of this prospective observational study was to assess the validity of the baseline Barthel Index score as a predictor of in-hospital mortality among 312 hospitalised COVID-19 adult patients. Based on it, the authors found a strong association between baseline patient functional status and in-hospital mortality due to COVID-19.

Please find below my comments for the authors:

Introduction

- Row 47, please use capital letter – Covid-19;

Author: Thank you very much for your comment. We have amended it to make it correct.

Materials and Methods

2.2. Sample/Participants

- Rows 69-71 - Patients were excluded if they had a positive PCR (Polymerase Chain Reaction) test or were positive for antigens, but did not have a radiological diagnosis of COVID-19. I suggest the authors to reformulate this as there is unclear if they excluded from the analysis the patients without Covid-19 pneumonia or the patients that were not evaluated by a radiologist. Was a chest X-ray or a CT scan used for the diagnosis of pneumonia? In addition, the basal Barthel Index was calculated at hospital admission by interview with questions about the patient’s stable situation prior to the disease. If the patients without pneumonia were excluded, did the authors considered in calculating the score the exact time until symptoms onset, before having Covid-19 pneumonia? Mild disease could also impacts this score, before the immune pase:

Use of the Barthel Index to Assess Activities of Daily Living before and after SARS-COVID 19 Infection of Institutionalized Nursing Home Patients. Int J Environ Res Public Health. 2021 Jul 7;18(14):7258. doi: 10.3390/ijerph18147258.

Measures of physical performance in COVID-19 patients: a mapping review. Pulmonology. 2021 NovDec;27(6):518-528. doi: 10.1016/j.pulmoe.2021.06.005.

Moreover, if patients without pneumonia were excluded, in this case the mortality would be assessed only for moderate disease, not for all patients.

Author: We agree that this section is confusing. We have proceeded to modify it. We would like to add that all patients underwent chest X-rays reported by a radiologist on admission. On the other hand, all patients analysed were diagnosed with COVID19 pneumonia. Other types of pneumonia not related to COVID19 were not included.

To clarify the exclusion criteria, we would like to add that in figure 3, we can see that 3 patients were excluded. In these cases, the patients were admitted for a traumatological problem. In the initial screening they presented positive PCR, but at no time did they show radiological signs of pneumonia.

On the other hand, we are grateful for the bibliographical references provided, which we find interesting. Despite this, as commented by reviewer 1, the Barthel index was calculated by asking about the patient's stable condition prior to infection with COVID19. For the sake of clarity, the text has been modified to better express it

2.3.3. Variables

- Did the authors recorded imaging findings for all Covid-19 patients? Was a chest X-ray or a CT scan used for the diagnosis of pneumonia?

Author: Yes, only 3 patients were excluded because they didn´t have a radiological diagnosis (Figure 1). All patients underwent a chest X-ray or a CT scan to stablish a diagnosis.

2.3.1. Primary Outcome

- The primary outcome was to identify the incidence of the in-hospital mortality among adult Covid-19 patients. Was it the all-cause mortality?

Author: No, the mortality studied was that attributed to COVID 19.

- When calculating PSI and CURB-65, were these scores applied only for patients with radiological pneumonia?

Author: All patients included in the study had their PSI and CURB-65 calculated. Furthermore, all patients were diagnosed with COVID-19 pneumonia. It should be noted that these instruments are validated for these situations (Fan G, 2020).

References:

Fan, G.; Tu, C.; Zhou, F.; Liu, Z.; Wang, Y.; Song, B.; Gu, X.; Wang, Y.; Wei, Y.; Li, H.; et al. Comparison of Severity Scores for COVID-19 Patients with Pneumonia: A Retrospective Study. Eur Respir J 2020, 56, 2002113, doi:10.1183/13993003.02113-2020

According to Table 1, the mean oxygen saturation was 91%, but this could also be due to other causes than Covid-19 pneumonia. Was it recorded as peripheral or by arterial samples for all patients?

Author: It is true that peripheral oxygen saturation can be affected by other causes. In our study the main diagnostic problem was COVID-19 pneumonia. We assume that the saturation alterations are mainly due to this problem. On the other hand, at the time peripheral oxygen saturation was obtained, arterial blood gases were drawn. The values of both measurements show a good correlation.

- Table 1. All these variables were recorded at hospital admission?

Author: Yes, except days of hospital stay.

What happened with deceased patients? Were they admitted in the ICU? ICU admission rates also have an important impact on primary outcome as it influence the all-cause mortality rates.

Author: Yes, some of them were admitted in ICU but we didn´t record that data as a variable. The patients were recruited in our study during the second semester after the establishment of the pandemic situation. At this time, various respiratory therapies were used in conventional wards and therefore only patients with an indication for mechanical ventilation were transferred to the ICU. These patients were mainly younger and had a good baseline functional situation. Even though it is highly probable that admission to the ICU could have a bias or impact on mortality, our main result was adjusted by age in our multivariate models number 2 and 3. This adjustment may reduce the impact of the question to which you refer.

2

2.5. Data analysis

- Quantitative variables were expressed as mean and standard deviation for normal distributions and median and interquartile range for non-normal distributions, and groups were compared using Student’s t-test. Did the authors checked the normal distribution of the variables in order to decide between t-test and Mann-Whitney U test?

Author: Yes, before performing the tests, the normality of the variables was analyzed. Subsequently, the most appropriate tests were used according to their distribution. We have added this issue in the statistical methods section.

(lines 136-137) “…comparisons. Normality was tested using the Kolmogorov-Smirnov test. Quantitative variables…”

(lines 139-140: …groups were compared using Student’s t-test or Mann-Whitney test as appropriate…)

3.1. Patient characteristics

- Rows 158-159 – the median comorbidity was 1. Is this the median values of the Charlson Comorbidity Index?

Author: We believe you are referring to what is described in rows 168-169. If so, you are correct, you are referring to the Charlson comorbidity Index Score. To avoid confusion, the sentence is reworded (Line 172).

- The most prevalent antecedent disease was hypertension. I suggest the authors to reformulate this as the term antecedent disease could be confusing. Comorbidity/chronic disease could be some terms that are more appropriate.

Author: Thank you very much for your suggestion, we agree with it. We have changed it. (Line 172).

Discussions

- Mortality in our study was 16.4%. Is this the all-cause mortality?

Author: No, the percentage of mortality expressed in the study refers to mortality attributed to COVID19 pneumonia infection.

- Authors could expand their discussion to more scientific data from literature. Other studies involving the relanshionship between the Barthel Index and outcomes in Covid-19 could be addressed. Here are some suggestion:

Barthel's Index: A Better Predictor for COVID-19 Mortality Than Comorbidities. Tuberc Respir Dis (Seoul). 2022 Oct;85(4):349-357. doi: 10.4046/trd.2022.0006. The prognostic value of the

Barthel Index for mortality in patients with COVID-19: A cross-sectional study. Front Public Health. 2023 Jan 24;10:978237. doi: 10.3389/fpubh.2022.978237.

Author: The suggestions are welcome. The articles have been reviewed and incorporated into the discussion.

Reviewer 3 Report

In this well-written article, the authors present to us the utility of the Bartel scale as a predictor of COVID-19 mortality. The article aims to use non-invasive scales as a classification tool for patients' severity of risk. As its strengths, it highlights the comparison against scales already used for the classification of mortality risk in respiratory infection, such as CURB-65 and PSI. However, I was a bit underwhelmed by the authors' strong conclusions. The scale using Bartel does not substantially improve what already exists, but the article can still be somewhat useful.

I have a couple of major suggestions to improve the potential of the article and a couple of minor considerations.

Major suggestion.

- There is no mention of the vaccination status of the patients. This point is very important because it can be a differential factor in the prognosis of patients. Therefore, I believe that the authors should include vaccination data, such as the type of vaccine, number of doses, and time since immunization. Without vaccination data, the article would not provide any contribution since currently the entire population is vaccinated.

- The other major consideration is to include the factor of time in the study of mortality. By this, I mean that the most accurate statistical method for modeling mortality is Cox regression, rather than logistic regression.

Minor suggestion

- You should change their discussion. Your results are useful, but the Bartel scale is not a better method. In fact, in these situations, it is important to have higher sensitivity to identify and act more quickly on those patients with a higher risk of mortality, even if it means having some false positives.

- The best way to express the scores for CURB-65, Barthel, PSI, and A-Drop is with median and interquartile range, given the nature of the variables.

Author Response

Reviewer 3

In this well-written article, the authors present to us the utility of the Bartel scale as a predictor of COVID-19 mortality. The article aims to use non-invasive scales as a classification tool for patients' severity of risk. As its strengths, it highlights the comparison against scales already used for the classification of mortality risk in respiratory infection, such as CURB-65 and PSI. However, I was a bit underwhelmed by the authors' strong conclusions. The scale using Bartel does not substantially improve what already exists, but the article can still be somewhat useful.

I have a couple of major suggestions to improve the potential of the article and a couple of minor considerations.

Major suggestion.

- There is no mention of the vaccination status of the patients. This point is very important because it can be a differential factor in the prognosis of patients. Therefore, I believe that the authors should include vaccination data, such as the type of vaccine, number of doses, and time since immunization. Without vaccination data, the article would not provide any contribution since currently the entire population is vaccinated.

Author: We agree with your statement. The patients were recruited in our study during the second semester after the establishment of the pandemic situation. At this time, nobody in our setting was vaccinated, so the vaccination rate was 0. So, we couldn´t study the possible impact. However, we don’t think that nowadays the entire population is vaccinated, there still big gaps in some places in the world like Africa. Income is also a determining factor for different vaccination rates. We attached a graphic from Our World in Data where we can visualize these differences. Therefore, we insist that today our study can be useful in those places where there are still no high rates of vaccination against COVID-19. It could even be useful in the face of possible new SARS-CoV-2 strains with immunological escape.

We add a graph extracted from the “ourworldindata.org”, which includes the current COVID19 vaccination rate.

 - The other major consideration is to include the factor of time in the study of mortality. By this, I mean that the most accurate statistical method for modeling mortality is Cox regression, rather than logistic regression.

 Author: There are different studies that analyze the predictive capacity of Barthel index and the other scores using logistic regression as a statistical method:

  1. Satici, C.; Asim Demirkol, M.; Sargin Altunok, E.; Gursoy, B.; Alkan, M.; Kamat, S.; Demirok, B.; Dilsah Surmeli, C.; Calik, M.; Cavus, Z.; et al. Performance of Pneumonia Severity Index and CURB-65 in Predicting 30-Day Mortality in Patients with COVID-19. 2020, doi:10.1016/j.ijid.2020.06.038.
  2. Artero, A.; Madrazo, M.; Fernández-Garcés, M.; Muiño Miguez, A.; González García, A.; Crestelo Vieitez, A.; E, G.G.; Em, F.A.; M, G.G.; M, A.M.; et al. Severity Scores in COVID-19 Pneumonia: A Multicenter, Retrospective, Cohort Study. J Gen Intern Med 2021, doi:10.1007/s11606-021-06626-7.
  3. Laosa, O.; Pedraza, L.; Álvarez-Bustos, A.; Carnicero, J.A.; Rodriguez-Artalejo, F.; Rodriguez-Mañas, L. Rapid Assessment at Hospital Admission of Mortality Risk From COVID-19: The Role of Functional Status. Journal of the American Medical Directors Association 2020, doi:10.1016/j.jamda.2020.10.002.

Cox regressions are used in studies where the main objective is to find the difference between times to the occurrence of an event, in this case mortality. As can be read at the beginning of the abstract, the aim of our study was "To assess the validity of the baseline Barthel Index score as a predictor of in-hospital mortality among COVID-19 patients." Therefore, we believe that the occurrence or non-occurrence of the in-hospital event is more interesting than the exact time of the event. Another operational issue for which we chose logistic regression is the ease of obtaining AUC ROC, sensitivity and specificity of the different models. We believe that these metrics are difficult to understand from models generated by Cox regression, as the sensitivities and specificities will change over time. It could happen that a patient could be a false negative (alive) at a certain point in time and yet subsequently die from COVID-19 on admission. For all these considerations, we maintain logistic regression as the optimal statistical method for the creation of multivariate models in our study. We thank you for your suggestion and for sure we will be able to perform other types of analysis in the future.

pMinor suggestion

 - You should change their discussion. Your results are useful, but the Bartel scale is not a better method. In fact, in these situations, it is important to have higher sensitivity to identify and act more quickly on those patients with a higher risk of mortality, even if it means having some false positives.

Author: We agree with your argument. With our results we believe that although the Barthel index was not created to predict mortality, it has acceptable results.

Table 2 shows that the sensitivity of the Barthel index is the lowest of all instruments, yet it has the second-best score, after PSI, of LR+. This means that it generates fewer false positives.

- The best way to express the scores for CURB-65, Barthel, PSI, and A-Drop is with median and interquartile range, given the nature of the variables.

Author: Although some authors have used median and standard deviation for these variables (Heras E, 2020), we agree with their suggestion, we modified the values by median and interquartile range both in the table and in the text.

Reference:

Heras E, Garibaldi P, Boix M, Valero O, Castillo J, Curbelo Y, Gonzalez E, Mendoza O, Anglada M, Miralles JC, Llull P, Llovera R, Piqué JM. COVID-19 mortality risk factors in older people in a long-term care center. Eur Geriatr Med. 2021 Jun;12(3):601-607. doi: 10.1007/s41999-020-00432-w. Epub 2020 Nov 27.

Round 2

Reviewer 2 Report

I would like to congratulate the authors on their work and I hope that my comments helped them to improve this manuscript.

I consider the changes on the manuscript were done accordingly and the comments are fine. No major issues are unsolved, except for a minor suggestion.

For the paragraph below, I would add this issue as a limitation of the study as I still consider it important.

Author: Yes, some of them were admitted in ICU but we didn´t record that data as a variable. The patients were recruited in our study during the second semester after the establishment of the pandemic situation. At this time, various respiratory therapies were used in conventional wards and therefore only patients with an indication for mechanical ventilation were transferred to the ICU. These patients were mainly younger and had a good baseline functional situation. Even though it is highly probable that admission to the ICU could have a bias or impact on mortality, our main result was adjusted by age in our multivariate models number 2 and 3. This adjustment may reduce the impact of the question to which you refer.

Author Response

Thank you very much for your contribution. We have included this aspect in the manuscript (lines 389-395).